# BAYESIAN HYPERNETWORKS

## ABSTRACT

We propose Bayesian hypernetworks: a framework for approximate Bayesian inference in neural networks. A Bayesian hypernetwork $h$ is a neural network which learns to transform a simple noise distribution, $p(\epsilon) = \mathcal{N}(\mathbf{0}, \mathbf{I})$, to a distribution $q(\boldsymbol{\theta}) := q(h(\epsilon))$ over the parameters $\boldsymbol{\theta}$ of another neural network (the "primary network"). We train $q$ with variational inference, using an invertible $h$ to enable efficient estimation of the variational lower bound on the posterior $p(\boldsymbol{\theta}|\mathcal{D})$ via sampling. In contrast to most methods for Bayesian deep learning, Bayesian hypernets can represent a complex multimodal approximate posterior with correlations between parameters, while enabling cheap iid sampling of $q(\boldsymbol{\theta})$. In practice, Bayesian hypernets provide a better defense against adversarial examples than dropout, and also exhibit competitive performance on a suite of tasks which evaluate model uncertainty, including regularization, active learning, and anomaly detection.

## 1 INTRODUCTION

Simple and powerful techniques for Bayesian inference of deep neural networks' (DNNs) parameters have the potential to dramatically increase the scope of applications for deep learning techniques. In real-world applications, unanticipated mistakes may be costly and dangerous, whereas anticipating mistakes allows an agent to seek human guidance (as in active learning), engage safe default behavior (such as shutting down), or use a "reject option" in a classification context.

DNNs are typically trained to find the single most likely value of the parameters (the "MAP estimate"), but this approach neglects uncertainty about which parameters are the best ("parameter uncertainty"), which may translate into higher *predictive* uncertainty when likely parameter values yield highly confident but contradictory predictions. Conversely, Bayesian DNNs model the full posterior distribution of a model's parameters given the data, and thus provides better calibrated confidence estimates, with corresponding safety benefits (Gal & Ghahramani, 2016; Amodei et al., 2016).[1] Maintaining a distribution over parameters is also one of the most effective defenses against adversarial attacks (Carlini & Wagner, 2017).

Techniques for Bayesian DNNs are an active research topic. The most popular approach is variational inference (Blundell et al., 2015; Gal, 2016), which typically restricts the variational posterior to a simple family of distributions, for instance a factorial Gaussian (Blundell et al., 2015; Graves, 2011). Unfortunately, from a safety perspective, variational approximations tend to underestimate uncertainty, by heavily penalizing approximate distributions which place mass in regions where the true posterior has low density. This problem can be exacerbated by using a restricted family of posterior distribution; for instance a unimodal approximate posterior will generally only capture a single mode of the true posterior. With this in mind, we propose learning an extremely flexible and powerful posterior, parametrized by a DNN $h$, which we refer to as a Bayesian hypernetwork in reference to Ha et al. (2017).

A Bayesian hypernetwork (BHN) takes random noise $\epsilon \sim \mathcal{N}(\mathbf{0}, \mathbf{I})$ as input and outputs a sample from the approximate posterior $q(\boldsymbol{\theta})$ for another DNN of interest (the "primary network"). The key

---

[1]While Bayesian deep learning may capture parameter uncertainty, most approaches, including ours, emphatically do *not* capture uncertainty about which model is correct (e.g., neural net vs decision tree, etc.). Parameter uncertainty is often called "model uncertainty" in the literature, but we prefer our terminology because it emphasizes the existence of further uncertainty about model specification.

insight for building such a model is the use of an invertible hypernet, which enables Monte Carlo estimation of the entropy term $-\log q(\boldsymbol{\theta})$ in the variational inference training objective.

We begin the paper by reviewing previous work on Bayesian DNNs, and explaining the necessary components of our approach (Section 2). Then we explain how to compose these techniques to yield Bayesian hypernets, as well as design choices which make training BHNs efficient, stable and robust (Section 3). Finally, we present experiments which validate the expressivity of BHNs, and demonstrate their competitive performance across several tasks (Section 4).

## 2 RELATED WORK

We begin with an overview of prior work on Bayesian neural networks in Section 2.1 before discussing the specific components of our technique in Sections 2.2 and 2.3.

### 2.1 BAYESIAN DNNS

Bayesian DNNs have been studied since the 1990s (Neal, 1996; MacKay, 1994). For a thorough review, see Gal (2016). Broadly speaking, existing methods either 1) use Markov chain Monte Carlo (Welling & Teh, 2011; Neal, 1996), or 2) directly learn an approximate posterior distribution using (stochastic) variational inference (Graves, 2011; Gal & Ghahramani, 2016; Salimans et al., 2015; Blundell et al., 2015), expectation propagation (Hernandez-Lobato & Adams, 2015; Soudry et al., 2014), or $\alpha$-divergences (Li & Gal, 2017). We focus here on the most popular approach: variational inference.

Notable recent work in this area includes Gal & Ghahramani (2016), who interprets the popular dropout (Srivastava et al., 2014) algorithm as a variational inference method ("MC dropout"). This has the advantages of being simple to implement and allowing cheap samples from $q(\boldsymbol{\theta})$. Kingma et al. (2015) emulates Gaussian dropout, but yields a unimodal approximate posterior, and does not allow arbitrary dependencies between the parameters.

The other important points of reference for our work are Bayes by Backprop (BbB) (Blundell et al., 2015), and multiplicative normalizing flows (Louizos & Welling, 2017). Bayes by Backprop can be viewed as a special instance of a Bayesian hypernet, where the hypernetwork only performs an element-wise scale and shift of the input noise (yielding a factorial Gaussian distribution).

More similar is the work of Louizos & Welling (2017), who propose and dismiss BHNs due to the issues of scaling BHNs to large primary networks, which we address in Section 3.3.[2] Instead, in their work, they use a hypernet to generate scaling factors, $\mathbf{z}$ on the means $\boldsymbol{\mu}$ of a factorial Gaussian distribution. Because $\mathbf{z}$ follows a complicated distribution, this forms a highly flexible approximate posterior: $q(\boldsymbol{\theta}) = \int q(\boldsymbol{\theta}|\mathbf{z})q(\mathbf{z})d\mathbf{z}$. However, this approach also requires them to introduce an auxiliary inference network to approximate $q(\mathbf{z}|\boldsymbol{\theta})$ in order to estimate the entropy term of the variational lower bound, resulting in lower bound *on the variational lower bound*.

Finally, the variational autoencoder (VAE) (Jimenez Rezende et al., 2014; Kingma & Welling, 2013) family of generative models is likely the best known application of variational inference in DNNs, but note that the VAE is *not* a Bayesian DNN in our sense. VAEs approximate the posterior over latent variables, given a datapoint; Bayesian DNNs approximate the posterior over model parameters, given a dataset.

### 2.2 HYPERNETWORKS

A hypernetwork (Ha et al., 2017; Brabandere et al., 2016; Bertinetto et al., 2016) is a neural net that outputs parameters of another neural net (the "primary network").[3] The hypernet and primary net together form a single model which is trained by backpropagation. The number of parameters of a DNN scales quadratically in the number of units per layer, meaning naively parametrizing a

---

[2] The idea is also explored by Shi et al. (2017), who likewise reject it in favor of their implicit approach which estimates the KL-divergence using a classifier.

[3] The name "hypernetwork" comes from Ha et al. (2017), who describe the general hypernet framework, but applications of this idea in convolutional networks were previously explored by Brabandere et al. (2016) and Bertinetto et al. (2016).

large primary net requires an impractically large hypernet. One method of addressing this challenge is Conditional Batch Norm (CBN) (Dumoulin et al., 2016), and the closely related Conditional Instance Normalization (CIN) (Huang & Belongie, 2017; Ulyanov et al., 2016), and Feature-wise Linear Modulation (FiLM) (Perez et al., 2017; Kirkpatrick et al., 2016), which can be viewed as specific forms of a hypernet. In these works, the weights of the primary net are parametrized directly, and the hypernet only outputs scale ($\gamma$) and shift ($\beta$) parameters for every neuron; this can be viewed as selecting which features are significant (scaling) or present (shifting). In our work, we employ the related technique of weight normalization (Salimans & Kingma, 2016), which normalizes the input weights for every neuron and introduces a separate parameter $g$ for their scale.

## 2.3 INVERTIBLE GENERATIVE MODELS

Our proposed Bayesian hypernetworks employ a differentiable directed generator network (DDGN) (Goodfellow et al., 2016) as a generative model of the primary net parameters. DDGNs use a neural net to transform simple noise (most commonly isotropic Gaussian) into samples from a complex distribution, and are a common component of modern deep generative models such as variational autoencoders (VAEs) (Kingma & Welling, 2013; Jimenez Rezende et al., 2014) and generative adversarial networks (GANs) (Goodfellow et al., 2014a; Goodfellow, 2017).

We take advantage of techniques for *invertible* DDGNs developed in several recent works on generative modeling (Dinh et al., 2014; 2016) and variational inference of latent variables (Rezende & Mohamed, 2015; Kingma et al., 2016). Training these models uses the change of variables formula, which involves computing the log-determinant of the inverse Jacobian of the generator network. This computation involves a potentially costly matrix determinant, and these works propose innovative architectures which reduce the cost of this operation but can still express complicated deformations, which are referred to as "normalizing flows".

## 3 METHODS

We now describe how variational inference is applied to Bayesian deep nets (Section 3.1), and how we compose the methods described in Sections 2.2 and 2.3 to produce Bayesian hypernets (Section 3.2).

## 3.1 VARIATIONAL INFERENCE

In variational inference, the goal is to maximize a lower bound on the marginal log-likelihood of the data, $\log p(\mathcal{D})$ under some statistical model. This involves both estimating parameters of the model, and approximating the posterior distribution over unobserved random variables (which may themselves also be parameters, e.g., as in the case of Bayesian DNNs). Let $\boldsymbol{\theta} \in \mathbb{R}^D$ be parameters given the Bayesian treatment as random variables, $\mathcal{D}$ a training set of observed data, and $q(\boldsymbol{\theta})$ a learned approximation to the true posterior $p(\boldsymbol{\theta}|\mathcal{D})$. Since the KL divergence is always non-negative, we have, for *any* $q(\boldsymbol{\theta})$:

$$\log p(\mathcal{D}) = \mathrm{KL}(q(\boldsymbol{\theta})\|p(\boldsymbol{\theta}|\mathcal{D})) + \mathbb{E}_q[\log p(\mathcal{D}|\boldsymbol{\theta}) + \log p(\boldsymbol{\theta}) - \log q(\boldsymbol{\theta})] \tag{1}$$
$$\geq \mathbb{E}_q[\log p(\mathcal{D}|\boldsymbol{\theta}) + \log p(\boldsymbol{\theta}) - \log q(\boldsymbol{\theta})]. \tag{2}$$

The right hand side of (2) is the evidence lower bound, or "ELBO".

The above derivation applies to any statistical model and any dataset. In our experiments, we focus on modeling conditional likelihoods $p(\mathcal{D}) = p(\mathcal{Y}|\mathcal{X})$. Using the conditional independence assumption, we further decompose $\log p(\mathcal{D}|\boldsymbol{\theta}) := \log p(\mathcal{Y}|\mathcal{X}, \boldsymbol{\theta})$ as $\sum_{i=1}^{n} \log p(\mathbf{y}_i|\mathbf{x}_i, \boldsymbol{\theta})$, and apply stochastic gradient methods for optimization.

## 3.1.1 VARIATIONAL INFERENCE FOR DEEP NETWORKS

Computing the expectation in (2) is generally intractable for deep nets, but can be estimated by Monte Carlo sampling. For a given value of $\theta$, $\log p(\mathcal{D}|\boldsymbol{\theta})$ and $\log p(\boldsymbol{\theta})$ can be computed and differentiated exactly as in a non-Bayesian DNN, allowing training by backpropagation. The entropy term $\mathbb{E}_q[-\log q(\boldsymbol{\theta})]$ is also straightforward to evaluate for simple families of approximate posteriors

such as Gaussians. Similarly, the likelihood of a test data-point under the predictive posterior using $S$ samples can be estimated using Monte Carlo:[4]

$$p(Y = y | X = x, \mathcal{D}) = \int p(Y = y | X = x, \boldsymbol{\theta}) p(\boldsymbol{\theta} | \mathcal{D}) d\boldsymbol{\theta} \tag{3}$$

$$\approx \frac{1}{S} \sum_{s=1}^{S} p(Y = y | X = x, \boldsymbol{\theta}_s), \quad \boldsymbol{\theta}_s \sim q(\boldsymbol{\theta}). \tag{4}$$

### 3.2 BAYESIAN HYPERNETS

Bayesian hypernets (BHNs) express a flexible $q(\boldsymbol{\theta})$ by using a DDGN (section 2.3), $h \in \mathbb{R}^D \to \mathbb{R}^D$, to transform random noise $\boldsymbol{\epsilon} \sim \mathcal{N}(\mathbf{0}, \mathbf{I}_D)$ into independent samples from $q(\boldsymbol{\theta})$. This makes it cheap to compute Monte Carlo estimations of expectations with respect to $q$; these include the ELBO, and its derivatives, which can be backpropagated to train the hypernet $h$.

Since BHNs are both trained and evaluated via samples of $q(\boldsymbol{\theta})$, expressing $q(\boldsymbol{\theta})$ as a generative model is a natural strategy. However, while DDGNs are convenient to sample from, computing the entropy term ($\mathbb{E}_q[-\log q(\boldsymbol{\theta})]$) of the ELBO additionally requires evaluating the likelihood of generated samples, and most popular DDGNs (such as VAEs and GANs) do not provide a convenient way of doing so.[5] In general, these models can be many-to-one mappings, and computing the likelihood of a given parameter value requires integrating over the latent noise variables $\boldsymbol{\epsilon} \in \mathbb{R}^D$:

$$q(\boldsymbol{\theta}) = \int q(\boldsymbol{\theta}; h(\boldsymbol{\epsilon})) q(\boldsymbol{\epsilon}) d\boldsymbol{\epsilon}. \tag{5}$$

To avoid this issue, we use an invertible $h$, allowing us to compute $q(\boldsymbol{\theta})$ simply by using the change of variables formula:

$$q(\boldsymbol{\theta}) = q_{\boldsymbol{\epsilon}}(\epsilon) \left| \det \frac{\partial h(\epsilon)}{\partial \boldsymbol{\epsilon}} \right|^{-1}, \tag{6}$$

where $q_{\boldsymbol{\epsilon}}$ is the distribution of $\boldsymbol{\epsilon}$ and $\boldsymbol{\theta} = h(\epsilon)$.

As discussed in Section 2.3, a number of techniques have been developed for efficiently training such invertible DDGNs. In this work, we employ both RealNVP (RNVP) (Dinh et al., 2016) and Inverse Autoregressive Flows (IAF) (Kingma et al., 2016). Note that the latter can be efficiently applied, since we only require the ability to evaluate likelihood of generated samples (not arbitrary points in the range of $h$, as in generative modeling applications, e.g., Dinh et al. (2016)); and this also means that we can use a lower-dimensional $\boldsymbol{\epsilon}$ to generate samples along a submanifold of the entire parameter space, as detailed below.

### 3.3 EFFICIENT PARAMETRIZATION AND TRAINING OF BAYESIAN HYPERNETS

In order to scale BHNs to large primary networks, we use the weight normalization reparametrization (Salimans & Kingma, 2016)[6]:

$$\boldsymbol{\theta}_j = g \, \mathbf{u}, \quad \mathbf{u} := \frac{\mathbf{v}}{\|\mathbf{v}\|_2}, \quad g \in \mathbb{R}, \tag{7}$$

where $\boldsymbol{\theta}_j$ are the input weights associated with a single unit $j$ in the primary network. We only output the scaling factors $g$ from the hypernet, and learn a maximum likelihood estimate of $\mathbf{v}$.[7] This allows us to overcome the computational limitations of naively-parametrized BHNs noted by Louizos & Welling (2017), since computation now scales linearly, instead of quadratically, in the number of

---

[4]Here we approximate the posterior distribution $p(\boldsymbol{\theta}|\mathcal{D})$ using the approximate posterior $q(\boldsymbol{\theta})$. We further use $S$ Monte Carlo samples to approximate the integral.

[5]Note that the entropy term is the only thing encouraging dispersion in $q$; the other two terms of (2) encourage the hypernet to ignore the noise inputs $\boldsymbol{\epsilon}$ and deterministically output the MAP-estimate for $\boldsymbol{\theta}$.

[6] Mathematical details can be found in the Appendix, Section B.

[7]This parametrization strongly resembles the "correlated" version of variational Gaussian dropout (Kingma et al., 2015, Sec. 3.2); the only difference is that we restrict the $\mathbf{u}$ to have norm 1.

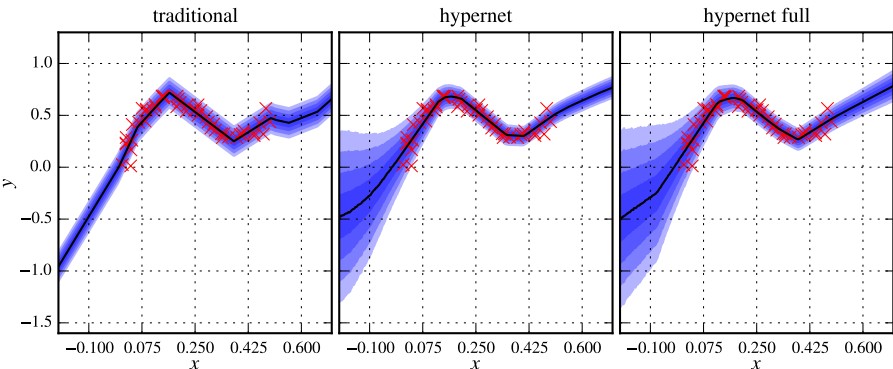

Figure 1: Illustration of BHNs (second and third) and a traditional non-Bayesian DNN (first) on the toy problem from Blundell et al. (2015). In the second subplot, we place a prior on the scaling factor $g$ and infer the posterior distribution using a BHN, while in the third subplot the hypernet is used to generate the whole weight matrices of the primary net. Each shaded region represents half a standard deviation in the posterior on the predictive mean. The red crosses are 50 examples from the training dataset.

primary net units. Using this parametrization restricts the family of approximate posteriors, but still allows for a high degree of multimodality and dependence between the parameters.

We also employ weight normalization within the hypernet, and found this stabilizes training dramatically. Initialization plays an important role as well; we recommend initializing the hypernet weights to small values to limit the impact of noise at the beginning of training. We also find clipping the outputs of the softmax to be within $(0.001, 0.999)$ critical for numerical stability.

## 4 EXPERIMENTS

We perform experiments on MNIST, CIFAR10, and a 1D regression task. There is no single metric for how well a model captures uncertainty; to evaluate our model, we perform experiments on regularization (Section 4.2), active learning (Section 4.3), anomaly detection (Section 4.4), and detection of adversarial examples (Section 4.5). Active learning and anomaly detection problems make natural use of uncertainty estimates: In anomaly detection, higher uncertainty indicates a likely anomaly. In active learning, higher uncertainty indicates a greater opportunity for learning. Parameter uncertainty also has regularization benefits: integrating over the posterior creates an implicit ensemble. Intuitively, when the most likely hypothesis predicts "A", but the posterior places more total mass on hypotheses predicting "B", we prefer predicting "B". By improving our estimate of the posterior, we more accurately weigh the evidence for different hypotheses. Adversarial examples are an especially difficult kind of anomaly designed to fool a classifier, and finding effective defenses against adversarial attacks remains an open challenge in deep learning.

For the hypernet architecture, we try both RealNVP (Dinh et al., 2016) and IAF(Kingma et al., 2016) with MADE(Germain et al., 2015), with 1-layer ReLU-MLP coupling functions with 200 hidden units (each). In general, we find that IAF performs better. We use an isotropic standard normal prior on the scaling factors ($g$) of the weights of the network. We also use Adam with default hyper-parameter settings (Kingma & Ba, 2014) and gradient clipping in all of our experiments. Our mini-batch size is 128, and to reduce computation, we use the same noise-sample (and thus the same primary net parameters) for all examples in a mini-batch. We experimented with independent noise, but did not notice any benefit. Our baselines for comparison are Bayes by Backprop (BbB) (Blundell et al., 2015), MC dropout (MCdropout) (Gal & Ghahramani, 2016), and non-Bayesian DNN baselines (with and without dropout).

### 4.1 QUALITATIVE RESULTS AND VISUALIZATION

We first demonstrate the behavior of the network on the toy 1D-regression problem from Blundell et al. (2015) in Figure 1. As expected, the uncertainty of the network increases away from the

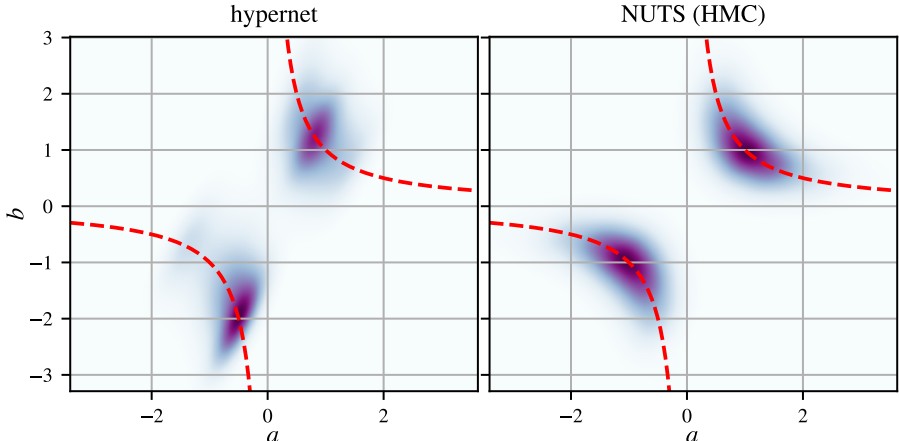

Figure 2: Learning the identity function with an overparametrized network: $\hat{y} = a \cdot b \cdot x$. This parametrization results in symmetries shown by the dashed red lines, and the Bayesian hypernetwork assigns significant probability mass to both modes of the posterior ($a = b = 1$ and $a = b = -1$).

observed data. We also use this experiment to evaluate the effects of our proposal for scaling BHNs via the weight norm parametrization (Section 3.3) by comparing with a model which generates the full set of parameters, and find that the two models produce very similar results, suggesting that our proposed method strikes a good balance between scalability and expressiveness.

Next, we demonstrate the distinctive ability of Bayesian hypernets to learn multi-modal, dependent distributions. Figure 6 (appendix) shows that BHNs do learn approximate posteriors with dependence between different parameters, as measured by the Pearson correlation coefficient. Meanwhile, Figure 2 shows that BHNs are capable of learning multimodal posteriors. For this experiment, we trained an over-parametrized linear (primary) network: $\hat{y} = a \cdot b \cdot x$ on a dataset generated as $y = x + \epsilon$, and the BHN learns capture both the modes of $a = b = 1$ and $a = b = -1$.

## 4.2 CLASSIFICATION

We now show that BHNs act as a regularizer, outperforming dropout and traditional mean field (BbB). Results are presented in Table 1. In our experiments, we find that BHNs perform on par with dropout on full datasets of MNIST and CIFAR10; furthermore, increasing the flexibility of the posterior by adding more coupling layers improves performance, especially compared with models with 0 coupling layers, which cannot model dependencies between the parameters. We also evaluate on a subset of MNIST (the first 5,000 examples); results are presented in the last two columns of Table 1. Replicating these experiments (with 8 coupling layers) for 10 trials yields Figure 3.

In these MNIST experiments, we use MLPs with 2 hidden layers of 800 or 1200 hidden units each. For CIFAR10, we train a convolutional neural net (CNN) with 4 hidden layers of $[64, 64, 128, 128]$ channels, $2 \times 2$ max pooling after the second and the fourth layers, filter size of 3, and a single fully connected layer of 512 units.

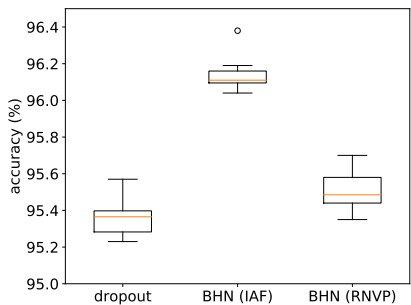

Figure 3: Box plot of performance across 10 trials. Bayesian hypernets (BHNs) with inverse autoregressive flows (IAF) consistently outperform the other methods.

Table 1: Generalization results on MNIST and CIFAR10 for BHNs with different numbers of RealNVP coupling layers (#), and comparison methods (dropout / maximum likelihood (MLE)). Bayes-by-backprop (Blundell et al., 2015) (*) models each parameter as an independent Gaussian, which is equivalent to using a hypernet with 0 coupling layers. We achieved a better result outputting a distribution over scaling factors (only). MNIST 5000 (A) and (B) are generalization results on subset (5,000 training data) of MNIST, (A) MLP with 800 hidden nodes. (B) MLP with 1,200 hidden nodes.

| MNIST 50,000 | | CIFAR10 50,000 | | MNIST 5,000 (A) | | MNIST 5,000 (B) | |
|---|---|---|---|---|---|---|---|
| # | Accuracy | # | Accuracy | # | Accuracy | # | Accuracy |
| 0 | 98.28% (98.01%*) | 0 | 67.83% | 0 | 92.06% | 0 | 90.91% |
| 2 | 98.39% | 4 | 74.77% | 8 | 94.25% | 8 | 96.27% |
| 4 | 98.47% | 8 | **74.90%** | 12 | **96.16%** | 12 | **96.51%** |
| 6 | 98.59% | dropout | 74.08% | dropout | 95.58% | dropout | 95.52% |
| 8 | 98.63% | MLE | 72.75% | | | | |
| dropout | **98.73%** | | | | | | |

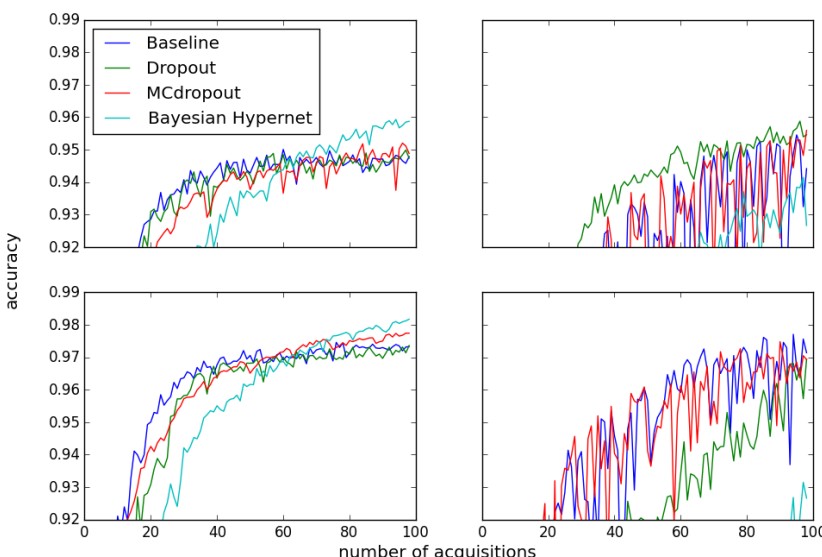

Figure 4: Active learning: Bayesian hypernets outperform other approaches after sufficient acquisitions when warm-starting (left), for both random acquisition function (top) and BALD acquisition function (bottom). Warm-starting improves stability for all methods, but hurts performance for other approaches, compared with randomly re-initializing parameters as in Gal et al. (2017) (right). We also note that the baseline model (no dropout) is competitive with MCdropout, and outperforms the Dropout baseline used by (Gal et al., 2017).[9] These curves are the average of three experiments.

## 4.3 ACTIVE LEARNING

We now turn to active learning, where we compare to the MNIST experiments of Gal et al. (2017), replicating their architecture and training procedure. Briefly, they use an initial dataset of 20 examples (2 from each class), and acquire 10 new examples at a time, training for 50 epochs between each acquisition. While Gal et al. (2017) re-initialize the network after every acquisition, we found that "warm-starting" from the current learned parameters was essential for good performance with BHNs, although it is likely that longer training or better initialization schemes could perform the same role. Overall, warm-started BHNs suffered at the beginning of training, but outperformed all other methods for moderate to large numbers of acquisitions.

---

[9]For the deterministic baseline, the value of the BALD acquisition function is always zero, and so acquisitions should be random, but due to numerical instability this is not the case in our implementation; surprisingly,

## 4.4 Anomaly Detection

Table 2: Anomaly detection on MNIST. Since we use the same datasets as Hendrycks & Gimpel (2016), we have the same base error rates, and refer the reader to that work.

| Dataset | MLP | | | MC dropout | | | BHN | | |
|---|---|---|---|---|---|---|---|---|---|
| | ROC | PR(+) | PR(−) | ROC | PR(+) | PR(−) | ROC | PR(+) | PR(−) |
| Uniform | 96.99 | 97.99 | 94.71 | 98.90 | 99.15 | 98.63 | 98.97 | 99.27 | 98.52 |
| OmniGlot | 94.92 | 95.63 | 93.85 | 95.87 | 96.44 | 94.84 | 94.89 | 95.56 | 93.64 |
| CIFARbw | 95.55 | 96.47 | 93.72 | 98.70 | 98.98 | 98.39 | 96.63 | 97.25 | 95.78 |
| Gaussian | 87.70 | 87.66 | 88.05 | 97.70 | 98.11 | 96.94 | 89.22 | 86.62 | 89.85 |
| notMNIST | 81.12 | 97.56 | 39.70 | 97.78 | 99.78 | 78.53 | 90.07 | 98.51 | 56.59 |

For anomaly detection, we take Hendrycks & Gimpel (2016) as a starting point, and perform the same suite of MNIST experiments, evaluating the ability of networks to determine whether an input came from their training distribution ("Out of distribution detection"). Hendrycks & Gimpel (2016) found that the confidence expressed in the softmax probabilities of a (non-Bayesian) DNN trained on a single dataset provides a good signal for both of these detection problems. We demonstrate that Bayesian DNNs outperform their non-Bayesian counterparts.

Just as in active learning, in anomaly detection, we use MC to estimate the predictive posterior, and use this to score datapoints. For active learning, we would generally like to acquire points where there is higher uncertainty. In a well-calibrated model, these points are also likely to be challenging or anomalous examples, and thus acquisition functions from the active learning literature are good candidates for scoring anomalies.

We consider all of the acquisition functions listed in (Gal et al., 2017) as possible scores for the Area Under the Curve (AUC) of Precision-Recall (PR) and Receiver Operating Characteristic (ROC) metrics, but found that the maximum confidence of the softmax probabilities (i.e., "variation ratio") acquisition function used by Hendrycks & Gimpel (2016) gave the best performance. Both BHN and MCdropout achieve significant performance gains over the non-Bayesian baseline, and MCdropout performs significantly better than BHN in this task. Results are presented in Table 2.

Second, we follow the same experimental setup, using all the acquisition functions, and exclude one class in the training set of MNIST at a time. We take the excluded class of the training data as out-of-distribution samples. The result is presented in Table 3 (Appendix). This experiment shows the benefit of using scores that reflect dispersion in the posterior samples (such as mean standard deviation and BALD value) in Bayesian DNNs.

## 4.5 Adversary Detection

Finally, we consider this same anomaly detection procedure as a novel tool for detecting adversarial examples. Our setup is similar to Li & Gal (2017) and Louizos & Welling (2017), where it is shown that when more perturbation is added to the data, model uncertainty increases and then drops. We use the Fast Gradient Sign method (FGS) (Goodfellow et al., 2014b) for adversarial attack, and use one sample of our model to estimate the gradient.[10] We find that, compared with dropout, BHNs are less confident on data points which are far from the data manifold. In particular, BHNs constructed with IAF consistently outperform RealNVP-BHNs and dropout in detecting adversarial examples and errors. Results are shown in Figure 5.

## 5 Conclusions

We introduce Bayesian hypernets (BHNs), a new method for variational Bayesian deep learning which uses an invertible hypernetwork as a generative model of parameters. BHNs feature efficient

---

we found the BALD values our implementation computes provide a better-than-random acquisition function (compare the blue line in the top and bottom plots).

[10] Li & Gal (2017) and Louizos & Welling (2017) used 10 and 1 model samples, respectively, to estimate gradient. We report the result with 1 sample; results with more samples are given in the appendix.

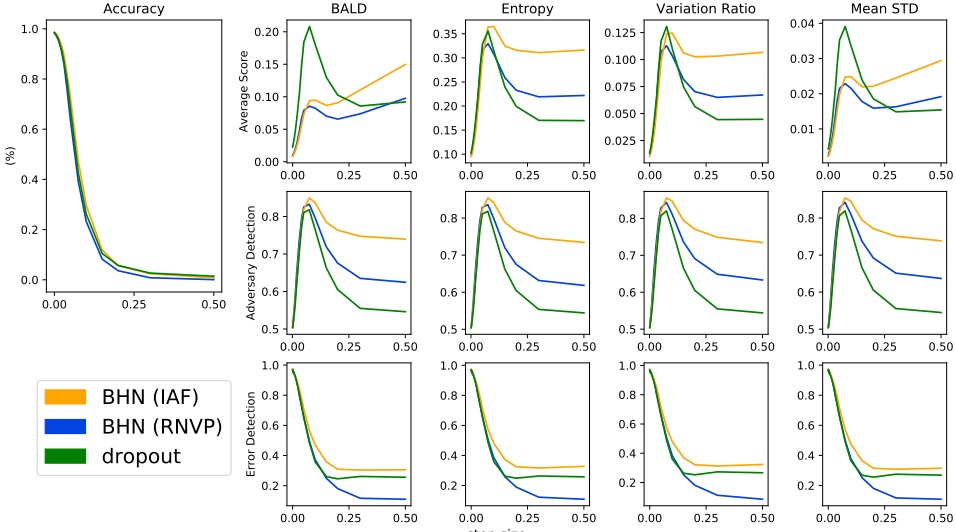

Figure 5: Adversary detection: Horizontal axis is the step size of the FGS algorithm. While accuracy drops when more perturbation is added to the data (left), uncertainty measures also increase (first row). In particular, the BALD and Mean STD scores, which measure epistemic uncertainty, are strongly increasing for BHNs, but *not* for dropout. The second row and third row plots show results for adversary detection and error detection (respectively) in terms of the AUC of ROC ($y$-axis) with increasing perturbation along the $x$-axis. Gradient direction is estimated with one Monte Carlo sample of the weights/dropout mask.

training and sampling, and can express complicated multimodal distributions, thereby addressing issues of overconfidence present in simpler variational approximations. We present a method of parametrizing BHNs which allows them to scale successfully to real world tasks, and show that BHNs can offer significant benefits over simpler methods for Bayesian deep learning. Future work could explore other methods of parametrizing BHNs, for instance using the same hypernet to output different subsets of the primary net parameters.

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

# A  ADDITIONAL RESULTS

## A.1  LEARNING CORRELATED WEIGHTS

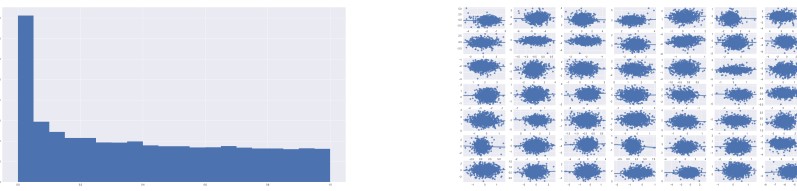

Figure 6: Histogram of Pearson correlation coefficient p-values (left) and a scatter matrix (right) of samples from a hypernet approximate posterior. We see that the hypernet posterior includes correlations between different parameters. Many of the p-values of the Pearson correlation test are below .05.

## A.2  UNSEEN MODE DETECTION

We replicate the experiments of anomaly detection with unseen classes of MNIST.

Table 3: Anomaly detection on MNIST with unseen classes. The first column indicates the missing class label in the training set. Top-most block: ROC score; middle: positive precision-recall; bottom: negative precision-recall.

|  | | Variation ratio | | | Mean std | | | BALD | | |
|---|---|---|---|---|---|---|---|---|---|---|
|  | MLP | dropout | BHN 4 | BHN 8 | dropout | BHN 4 | BHN 8 | dropout | BHN 4 | BHN 8 |
| 0 | 95.52 | 97.44 | 96.62 | 96.45 | **97.90** | 96.53 | 96.77 | 97.89 | 96.59 | 96.55 |
| 1 | 96.70 | 94.60 | 96.62 | 96.46 | 94.01 | 96.62 | 96.25 | 93.92 | **96.92** | 96.19 |
| 2 | 92.83 | 95.77 | 92.99 | 93.47 | 96.02 | 93.03 | 93.57 | **96.08** | 93.59 | 94.26 |
| 3 | 93.03 | 93.11 | 95.03 | **95.34** | 93.65 | 94.86 | 94.77 | 93.65 | 94.87 | 94.96 |
| 4 | 89.08 | 88.96 | 75.73 | 81.19 | **89.45** | 75.73 | 81.31 | 89.34 | 74.31 | 84.34 |
| 5 | 88.53 | 94.66 | 93.20 | 87.95 | 95.37 | 93.08 | 88.31 | **95.45** | 92.61 | 85.77 |
| 6 | 95.40 | 96.33 | 93.67 | 94.69 | **96.99** | 93.80 | 94.80 | 96.96 | 93.27 | 94.50 |
| 7 | 92.46 | 96.61 | 95.08 | 93.70 | **97.08** | 94.68 | 92.82 | 97.06 | 94.88 | 92.89 |
| 8 | 96.35 | **98.05** | 95.86 | 96.85 | 97.67 | 95.74 | 96.98 | 97.23 | 95.48 | 96.87 |
| 9 | 94.75 | 95.95 | 95.62 | **96.54** | 96.03 | 95.46 | 96.42 | 96.10 | 95.84 | 96.37 |
| 0 | 97.68 | 98.68 | 98.34 | 98.32 | **98.87** | 98.31 | 98.45 | **98.87** | 98.35 | 98.35 |
| 1 | 98.26 | 97.03 | 98.23 | 98.15 | 96.58 | 98.20 | 98.04 | 96.58 | **98.35** | 98.00 |
| 2 | 96.06 | 97.74 | 95.63 | 96.07 | 97.83 | 95.31 | 96.01 | **97.87** | 95.80 | 96.45 |
| 3 | 96.00 | 95.74 | 97.28 | **97.68** | 95.97 | 97.09 | 97.37 | 96.00 | 97.13 | 97.49 |
| 4 | 93.73 | 93.93 | 84.66 | 86.40 | **94.10** | 85.16 | 86.46 | 94.00 | 83.32 | 90.00 |
| 5 | 93.92 | 97.31 | 96.79 | 93.15 | 97.60 | 96.62 | 93.34 | **97.61** | 96.34 | 90.72 |
| 6 | 97.68 | 97.99 | 96.38 | 97.27 | **98.29** | 96.55 | 97.29 | **98.29** | 96.05 | 97.13 |
| 7 | 95.56 | 98.16 | 97.40 | 96.51 | **98.36** | 97.07 | 95.82 | 98.32 | 97.17 | 95.89 |
| 8 | 98.18 | **99.03** | 97.97 | 98.37 | 98.87 | 97.96 | 98.53 | 98.70 | 97.83 | 98.45 |
| 9 | 97.32 | 97.94 | 97.76 | 98.27 | 97.93 | 97.71 | **98.31** | 98.02 | 98.00 | 98.29 |
| 0 | 90.11 | 94.44 | 92.17 | 90.95 | **96.08** | 92.06 | 92.67 | **96.08** | 91.92 | 91.91 |
| 1 | 92.84 | 89.08 | 92.48 | 91.99 | 88.11 | 92.71 | 91.53 | 87.67 | **93.11** | 91.55 |
| 2 | 85.74 | 91.13 | 86.61 | 87.52 | 92.04 | 88.22 | 88.51 | **92.16** | 89.20 | 89.90 |
| 3 | 87.46 | 87.78 | 89.46 | 88.75 | 89.72 | 89.99 | 87.09 | 89.78 | **90.22** | 87.21 |
| 4 | 80.96 | 79.04 | 64.02 | 72.11 | 81.82 | 64.33 | 73.69 | **81.89** | 64.16 | 75.72 |
| 5 | 80.41 | 87.74 | 84.15 | 78.16 | 90.48 | 84.85 | 78.96 | **90.81** | 84.27 | 76.99 |
| 6 | 89.34 | 92.26 | 88.17 | 88.60 | **94.21** | 88.28 | 89.10 | 94.07 | 87.14 | 87.89 |
| 7 | 87.08 | 92.69 | 88.91 | 86.85 | 94.02 | 89.07 | 86.64 | **94.33** | 89.70 | 86.71 |
| 8 | 91.88 | **95.82** | 90.52 | 92.83 | 94.40 | 89.69 | 92.82 | 92.80 | 88.41 | 92.08 |
| 9 | 88.10 | 90.71 | 89.70 | **91.67** | 91.49 | 88.72 | 90.85 | 91.56 | 88.79 | 90.30 |

### A.3 STRONGER ATTACK

Here we use 32 samples to estimate the gradient direction with respect to the input. A better estimate of gradient amounts to a stronger attack, so accuracy drops lower for a given step size while an adversarial example can be more easily detected with a more informative uncertainty measure.

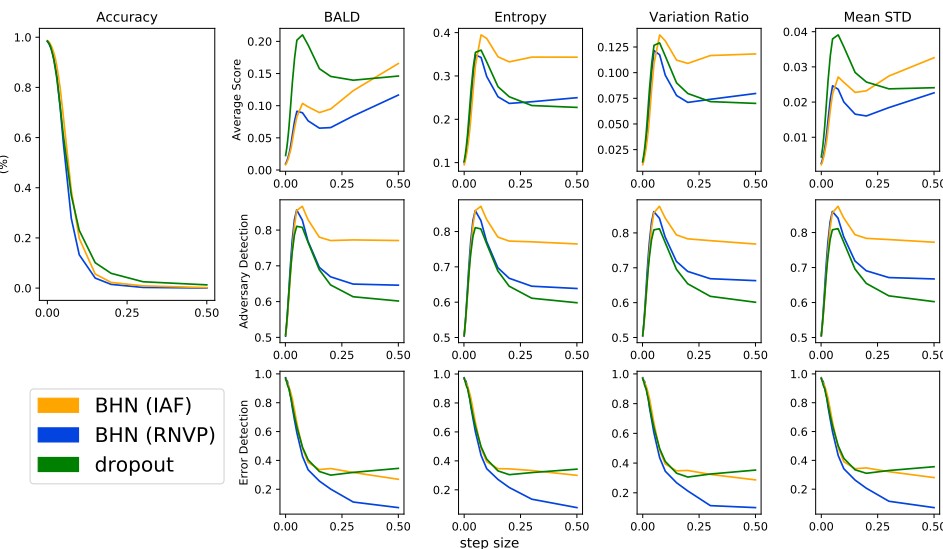

Figure 7: Adversary detection with 32-sample estimate of gradient.

## B DERIVATION OF TRAINING OBJECTIVE

In this paper, we employ weight normalization in the primary network (7), treating (only) the scaling factors $\mathbf{g}$ as random variables. We choose an isotropic Gaussian prior for $\mathbf{g}$: $p(\mathbf{g}) = \mathcal{N}(\mathbf{g}; \mathbf{0}, \lambda \mathbf{I})$, which results in an $L_2$ weight-decay penalty on $\mathbf{g}$, or, equivalently, $\mathbf{w} = \mathbf{g}\frac{\mathbf{v}}{||\mathbf{v}||_2}$. Our objective and lower bound are then:

$$\log p(\mathcal{D}; \mathbf{v}, \mathbf{b}) = \log \int_{\mathbf{g}} p(\mathcal{D}|\mathbf{g}; \mathbf{v}, \mathbf{b})p(\mathbf{g})d\mathbf{g} \tag{8}$$

$$\geq \mathbb{E}_{q(\mathbf{g})}[\log p(\mathcal{D}|\mathbf{g}; \mathbf{v}, \mathbf{b}) + \log p(\mathbf{g}) - \log q(\mathbf{g})] \tag{9}$$

$$\geq \mathbb{E}_{\boldsymbol{\epsilon} \sim q_{\boldsymbol{\epsilon}}(\boldsymbol{\epsilon}), \mathbf{g}=h_\phi(\boldsymbol{\epsilon})}[\log p(\mathcal{D}|\mathbf{g}; \mathbf{v}, \mathbf{b}) + \log p(\mathbf{g}) - \log q(\boldsymbol{\epsilon}) + \log \left|\det \frac{\partial h_\phi(\boldsymbol{\epsilon})}{\partial \boldsymbol{\epsilon}}\right|] \tag{10}$$

where $\mathbf{v}$ and $\mathbf{b}$ are the direction and bias parameters of the primary net, and $\phi$ is the parameters of the hypernetwork. We optimize this bound with respect to $\{\mathbf{v}, \mathbf{b}, \phi\}$ during training.

