# OpenReview forum: "Bayesian Hypernetworks"
_ICLR.cc/2018/Conference — Reject_

### Official Review · AnonReviewer2 · 2017-11-26
**Interesting paper - novel enough?**

**Rating:** 6
**Confidence:** 4

**Review:**

This paper proposes Bayesian hypernetworks to carry out Bayesian learning of deep networks. The idea is to construct a generative model capable of approximating the posterior distribution over the parameters of deep networks. I think that the paper is well written and easy to follow.

I like the idea of constructing general approximation strategies for complex posterior distribution and the proposed approach inherits all the scalability properties of modern deep learning techniques. In this respect, I think that the paper tackles a timely topic and is interesting to read.

It is not entirely clear to me why the Authors name their proposal Bayesian hypernetworks. This seems to suggest that also the hypernetwork is infered using Bayesian inference, but if I understand correctly this is not the case.

I have some comments on novelty and realization of the experiments. In the positioning of the work in the literature, the Authors point out that hypernetworks have been proposed before, so it is not clear what is the actual novelty in the proposal. Is it the use of Real NVPs and IAFs as hypernetworks? These methods have been already proposed and extensively studied in the literature, and even if they have been adapted to be hypernetworks here, I believe that the novelty is fairly limited.

The experimental part is interesting as it explores a number of learning scenarios. However, I think that it would have been useful to add comparisons with standard variational inference (e.g., Graves, 2011) for deep networks to substantiate the claims that this approach underestimates uncertainty. I believe that this would strengthen the comparative evaluation.

I think the paper would have made a stronger case by including other approaches to approximate posteriors using generative models. For example, the variational Gaussian process paper sounds like an ideal method to include here.

[1] D. Tran, R. Ranganath, and D. M. Blei. Variational Gaussian process. arXiv preprint arXiv:1511.06499, 2015.

---

> ### Public Comment · (anonymous) · 2018-01-05
> **Rebuttal**
>
> Thanks for the feedback; we’ll respond item-wise.
>
> “It is not entirely clear to me why the Authors name their proposal Bayesian hypernetworks. This seems to suggest that also the hypernetwork is infered using Bayesian inference, but if I understand correctly this is not the case.”
> That’s correct.  Rather, the hypernetwork *performs* (variational) Bayesian inference, which is why we chose this name.
>
>  “ [...] it is not clear what is the actual novelty in the proposal. Is it the use of Real NVPs and IAFs as hypernetworks?”
> The novelty of our approach (over e.g. “HyperNetworks” (Ha et al. 2016)  is not the architecture of the hypernet, but rather the idea of inputting noise to the hypernetwork (as opposed to learned parameters or network activations) in order to learn a distribution over network parameters.  But also note that other recent works (e.g. Luizos and Welling 2017) have also proposed similar ideas (without drawing connections to hypernetworks).
>
> ”However, I think that it would have been useful to add comparisons with standard variational inference (e.g., Graves, 2011) for deep networks”
> We do compare with standard mean-field Variational Inference as implemented by Blundell et al. (2015) in their paper “Weight Uncertainty in Neural Networks”, which we consider a stronger baseline than Graves (2011).  With 0 coupling layers, Bayesian Hypernetworks reduce to a naive mean file method, as before applying additional normalizing flows, the spherical Gaussian noise is transformed by an element-wise scale and shift layer, yielding an arbitrary diagonal Gaussian.
> The method of Blundell et al. (2015) is more modern and simpler than Graves’; Blundell et al. use the reparametrization trick to compute unbiased estimates of gradients of the variational posterior, whereas Graves uses Gaussian gradient identities due to the Bonnet and Price theorem, to estimate the gradient and further approximate the Fisher information matrix for the sigma parameters via a diagonal approximation of the Hessian.  This yields a biased and noisy estimate of gradient, and Graves’ method is typically outperformed by more modern approaches, see, for instance Hernandez-Lobato and Adams (2015).
>
> “I think the paper would have made a stronger case by including other approaches to approximate posteriors using generative models. For example, the variational Gaussian process paper sounds like an ideal method to include here.”
> We agree that comparing with more approaches would be valuable, but a thorough comparison of the many existing Bayesian methods is out of our scope.  In this work, we chose to compare with two of the most popular modern techniques for variational inference in neural networks, Bayes by Backprop (Blundell et al., 2015), and MCdropout (Gal and Ghahramani, 2016).  Is there any reason you believe variational Gaussian processes [1] in particular are an ideal method to compare with?
>
> References:
> - David Ha, Andrew Dai, and Quoc V. Le. Hypernetworks. 2017. URL https://openreview. net/pdf?id=rkpACe1lx.
> - Charles Blundell, Julien Cornebise, Koray Kavukcuoglu, and Daan Wierstra. Weight uncertainty in neural networks. In Proceedings of The 32nd International Conference on Machine Learning, pp. 1613–1622, 2015.
> - Christos Louizos and Max Welling. Multiplicative normalizing flows for variational bayesian neural networks. arXiv e-prints, March 2017.
> - Jose Miguel Hernandez-Lobato and Ryan Adams. Probabilistic backpropagation for scalable learning of Bayesian neural networks. In Proceedings of The 32nd International Conference on Machine Learning, pp. 1861–1869, 2015.
> - Yarin Gal and Zoubin Ghahramani. Dropout as a Bayesian approximation: Representing model uncertainty in deep learning. In International Conference on Machine Learning, pp. 1050–1059, 2016.

---

> > ### Comment · AnonReviewer2 · 2018-01-12
> > **Response to rebuttal**
> >
> > Many thanks for the detailed response. I still think that the novelty is limited and that a wider experimental work would have strengthen the paper.
> >
> > My reason to use variational GPs is that they would be an alternative way to generate a posterior over model parameters.
> >
> > I'll probably leave my score to a 6

---

### Official Review · AnonReviewer1 · 2017-11-27
**Incremental idea with a potential technical issue.**

**Rating:** 6
**Confidence:** 4

**Review:**

This paper presents Bayesian Hypernetworks; variational Bayesian neural networks where the variational posterior over the weights is governed by a hyper network that implements a normalizing flow (NF) such as RealNVP and IAF. As directly outputting the weight matrix with a hyper network is computationally expensive the authors instead propose to utilize weight normalisation on the weights and then use the hyper network to output scalar scaling variables for each hidden unit, similarly to what was done at [1]. The main difference with this prior work is that [1] consider these NF scaling variables as auxiliary random variables to a mean field Gaussian distribution over the weights whereas this paper attempts to posit a distribution directly on the weights via the NF. This avoids the nested variational approximation and auxiliary models of [1], which can potentially yield a tighter bound. The proposed method is evaluated on extensive experiments.

This paper seems like a plausible idea with extensive experiments but the similarity with [1] make it an incremental contribution and, furthermore, it seems that it has a technical issue with what is explained at Section 3.3. More specifically, if you generate the parameters \theta according to Eq. 7 and posit a prior over \theta then you will have a problematic variational bound as there will be a KL divergence, KL(q(\theta) || p(\theta)), with distributions of different support (since q(\theta) is defined only along the directions spanned by u), which is infinite. For the KL to be valid you will need to posit a prior distribution over `g`, p(g), and then consider KL(q(g) || p(g)), with q(g) being given by the NF. From the experiment paragraph at page 5 though I deduct that you instead employ “an isotropic standard normal prior over the weights”, i.e. \theta, thus I believe that you indeed have a problematic bound. How do you actually compute logq(\theta) when you employ the parametrisation discussed at 3.3? Did you use that parametrisation in every experiment?

Other than that, I believe that it would be interesting to experiment with a `full` hyper network, i.e. generating directly the entire parameter vector \theta, e.g. at the toy regression experiment where the dimensionality is small. This would then better illustrate the tradeoffs you make when you reduce the flexibility of the hyper-network to just outputting the row scaling variables and the effect this has at the posterior approximation.

Typos:
(1) Page 3, 3.1.1 log(\theta) -> logp(\theta).
(2) Eq. 6, it needs to be |det \frac{\partial h(\epsilon)}{\partial \epsilon}|^{-1} or |det \frac{\partial h^{-1}(\theta)}{\partial \theta}| for a valid change of variables formula.

[1] Louizos & Welling, Multiplicative Normalizing Flows for Variational Bayesian Neural Networks.

---

> ### Public Comment · (anonymous) · 2018-01-05
> **Rebuttal**
>
> Thanks for the thoughtful and detailed comments.
>
> You are correct that we must “posit a prior distribution over `g`, p(g), and then consider KL(q(g) || p(g)), with q(g) being given by the NF”, and this is in fact what we do.  We recognize that the original submission was misleading on this point, and we’ve clarified our approach in the main text and an additional section of the appendix.  Note that the log-likelihoods log p(theta) and log p(g) actually yield equivalent L2 weight-decay penalty terms (under a spherical Gaussian prior), and this justifies our choice of prior on g as a means of encouraging simpler models.
>
> To compute log q(theta), we use the change of variables equation from equation 6 (with theta replaced with g using the weight norm parametrization); since we only need to evaluate log q(g) (and derivatives) at sampled values of g, IAF or RNVP are both able to compute the determinants efficiently.
>
> We did use the weight-norm parametrization of section 3.3 in all of the experiments, but following your suggestion, we’ve included results with a fully-parametrized Bayesian Hypernetwork in Figure 1 of the updates paper.  The results are nearly identical, indicating that our method makes a good trade-off between performance and scalable computation.
>
> We’ve also corrected the typos; thanks for catching them!

---

> > ### Comment · AnonReviewer1 · 2018-01-12
> > **Response to rebuttal**
> >
> > Thank you for the clarifications; I have revised the score of the paper according to your comments. It is interesting that the quality of the predictive distribution does not change for the toy task when employing a full hypernetwork and probably points at optimization difficulties for the bound (as the uncertainty still does not increase for the right part of the function).

---

### Official Review · AnonReviewer3 · 2017-11-27
**interesting idea. not rigorous. limited novelty**

**Rating:** 6
**Confidence:** 4

**Review:**

* Edit: I increased my rating to 6. The authors fixed the first error I pointed out below. Regarding the second point: I still think it is possible to take a limit of sigma -> 0 in MNF, which makes the methods very similar.

The authors propose a new method of defining approximate posteriors for use in Bayesian neural networks. The idea of using hypernetworks for Bayesian inference is compelling, and the authors show some promising first results. I see two issues, and would be willing to increase my rating if these were sufficiently addressed.

- The paper says it uses an "isotropic standard normal prior on the weights of the network". However, the stochastic part of the generated weights (i.e. the scales) is of a lower dimension than the weights. It seems to me this means that the KL divergence between prior and posterior is undefined, or infinite, as the posterior is only defined on a sub-manifold. What exactly is the loss term that is added to the training objective? And how is this justified?

- The instantiation of Bayesian hypernetworks that is used in experiments seems to be a special case of the method of multiplicative normalizing flows as proposed by Louizos and Welling and discussed in this paper. If the variances / sigmas are zero in the latter method, their approximation seems functionally equivalent to Bayesian hypernetworks (though with different parameterization). Is my understanding correct? If so, the novelty of the proposed method is limited.

---

> ### Public Comment · (anonymous) · 2018-01-05
> **Rebuttal**
>
> Thank you for the helpful feedback.  We’ll aim to address both issues in a satisfying way.
>
> First, you are correct that the manifold of generated weights is of lower dimensionality than the space of all possible weights.
> In fact, the quoted "isotropic standard normal prior on the weights of the network" in the experiments section should have said “[...] prior on the scaling factors (g) of the weights of the network”, and we’ve made that correction in the updated paper.  Thus there is no technical issue with computing the KL-divergence, although this also means that we only maintain uncertainty over g, and use point estimates for the directions of the weights.  Note, however, that the scaling factors exactly control the norm of the weights, and so our prior distribution on g expresses a preference for simpler (smaller-norm) weights and lower-complexity models, just as a prior on the weights (theta) would; this justifies our choice of this prior.
>
> Second, our method is not actually a special case of multiplicative normalizing flows (MNF), since the derivation of MNF prohibits reducing the variance to 0.  Our methods *are* quite similar, but make different trade-offs in order to allow scaling to large networks.  Mathematically, MNF treats all the neural net parameters as random variables, and derives a lower-bound on the ELBO to allow for a hierarchical posterior (z->W); whereas we only treat g as random variables, and perform standard variational inference on this model.
> Further differences in our work are:
> 1) While the scaling outputs of the normalizing flow in MNF (z) operate on units activations (or equivalently, outgoing weights), in BHNs, the scaling factors (g) operate on the pre-activations (or equivalently, incoming weights).
> 2) We normalize the direction component of the weights (u := v/||v||), which means that the scale of g (which is analogous to z, in MNF) can control the complexity of the model.  Crucially, this allows us to place a meaningful prior on g, and thus avoid introducing an auxiliary inference model for g.
> 3) We also perform a more broad experimental evaluation, including experiments on active learning.
>
> We hope this addresses your concerns, and are happy to continue the conversation otherwise.

---

> ### Public Comment · (anonymous) · 2018-01-26
> **Regarding Reviewer3's comparison with MNF (sigma --> 0)**
>
> We agree that *mechanically*, the procedure for sampling the posterior in MNF and BHN is very similar, to whit:
> 1. in BHNs, we sample the (scaling factors of the) parameters directly; this is equivalent to scaling units’ pre-activations.
> 2. in MNF, they sample z (which can be viewed as a scaling factor of the activations), and then add some i.i.d. Gaussian noise (with std-dev sigma) to the resulting parameters.
> So when sigma --> 0 in MFN, the only difference would be whether the outputs of the flow are used to rescale the activations or pre-activations.
>
> Nevertheless, we don’t think the derivation in the MNF paper would behave well mathematically as sigma --> 0.
> This sigma refers to the std-dev of q(W | zTf), and equation 13 of the paper (https://arxiv.org/pdf/1703.01961.pdf) includes the term −KL(q(W|zTf)||p(W)), which will go to -infinity as sigma --> 0 (since p(W) has a fixed variance).
> This seems problematic, since this term is part of the objective function of MNF (eqn7).
>
> So injecting extra i.i.d. Gaussian noise in the parameter space seems fundamental to MNF.  This could be disadvantageous, since i.i.d. noise might put probability mass in poor regions of parameter space, although we might also expect it to provide additional regularization benefits (as Gaussian dropout does).

---

### Comment · AnonReviewer3 · 2017-11-23
**some questions**

* You use the same noise sample for all examples in a minibatch. Is this because the computation in the hypernet otherwise becomes too expensive? An advantage of your proposed approach seems to be that by outputting only the scales of the parameters you could easily use different scales for different examples as far as the primary network is concerned.

* You claim to show "that BHNs act as a regularizer, outperforming dropout and traditional mean field". However the results shown in e.g. table 1 for CIFAR-10 seem to be quite a bit worse than previous SOTA results obtained with dropout. Why the difference?

* Please expand the caption in Figure 3. Is this MNIST? With the full training set or a restricted set?

* How do your anomaly detection results compare against methods that use ensembles? (e.g. http://papers.nips.cc/paper/7219-simple-and-scalable-predictive-uncertainty-estimation-using-deep-ensembles)

---

> ### Public Comment · (anonymous) · 2017-12-24
> **some answers**
>
> 1) We didn't experiment with it very much. In our preliminary investigations it didn't seem to make much difference, so we just didn't look into it more. This is consistent with Blundell et al’s observation (see Bayes by backprop paper sec 3.1 last paragraph).
>
> 2) We're using a different architecture. We just chose something that was easy to implement and train quickly. For instance, we don't use residual layers.
>
> 3) 5000 examples. This is mentioned in the text, but we can add it to the caption as well
>
> 4) We did not compare with ensemble methods.

---

### Public Comment · (anonymous) · 2017-11-27
**Missing literature in recent development of Bayesian Neural Networks**

Is it possible to compare with particle [1] and sample-based [2] methods to learn the weight uncertainty of neural networks? which have shown excellent performance.

[1] Stein Variational Gradient Descent: A General Purpose Bayesian Inference Algorithm, 2016
[2] Preconditioned Stochastic Gradient Langevin Dynamics for Deep Neural Networks, 2016

---

> ### Public Comment · (anonymous) · 2017-12-24
> **Our focus: Variational Inference (vs. MCMC and Particle based methods)**
>
> There are many recent works on Bayesian DNNs, and it’s beyond out scope to compare with all of them, although of course it would be possible to do so and interesting to see the results.
> The methods you mention have some significant differences from Variational methods (such as Bayesian Hypernets), with corresponding pros and cons, and thus somewhat different use cases. Thus we believe that improving approximate (e.g. Variational) inference approaches to Bayesian DNNs is a worthwhile research direction in and of itself.
> Particle-based methods are similar to ensemble methods: they train several different models in parallel. In [1], they try to maximize pairwise distances between N different models. This has the following pros and cons:
> Pros:
> •	More likely to express different modes (with up to N modes)
> Cons:
> •	Needs N time more memory.
> •	Needs N^2 computation for the pairwise distances.
> •	Only gives you access to N samples from the posterior.
> Meanwhile, Markov chain-based methods (such as SGLD [2] and precursors) have the advantage of allowing asymptotically unbiased samples from the true posterior (unlike particle-based or variational methods).  However, the Metropolis Hastings step of [2] is expensive since the whole dataset needs to be evaluated; as a result, this step is usually not performed in practice, resulting in additional bias. Furthermore, successive samples from such Markov chain-based methods are typically highly correlated. This autocorrelation introduces bias if the algorithm terminates prematurely. In contrast, variational inference methods that feature directed sampling of the approximate posterior (such as our work) yield independent samples by construction. The disadvantage of variational inference methods is that they can produce poor samples of model parameters when the true posterior is not well approximated by the family of approximating distributions; our work addresses this problem by using a more flexible approximate posterior.

---

> > ### Public Comment · (anonymous) · 2017-12-25
> > **Revise the title?**
> >
> > Thanks for clarifying the difference. Well done.
> >
> > Since the the paper focuses on the variational methods for DNNs using hypernetworks, is it possible to revise the title to better reflect the content? The current title gives the impression that the paper represents all Bayesian inference methods.

---

### Public Comment · (anonymous) · 2017-12-24
**Clarifying: we use KL(p(g) || q(g)), which is not degenerate.**

Reviewers 1 and 3 rightly note that our experiments section states that we place an isotropic Gaussian prior over the weights of the network.  We apologize for the confusion.  In fact, our prior and posterior are over the scaling factors, g, so the KL divergence is well-defined.  Specifically, we use an isotropic Gaussian prior for g.  The resulting penalty term is -log(p(g)) + log(q(g)).

Note that under the weight-norm parametrization, ||w|| == g, and in fact, the -log(p(g)) term and it’s gradients are equivalent to a weight-decay penalty on the sampled weights (w), and thus encourage smaller norm weights and lower complexity functions, just like an isotropic Gaussian prior on the weights would.

---

### Public Comment · (anonymous) · 2018-01-05
**Summary of Rebuttals and Paper Updates**

The reviewers’ main concerns were: 1) the technical soundness of our approach (specifically WRT the potential degeneracy of the KL-divergence), and 2) lack of novelty, especially with respect to the approach of Luizos and Welling [1].  The first concern is addressed in our comment “Clarifying: we use KL(p(g) || q(g)), so it’s not degenerate.”

We acknowledge the similarity with Multiplicative Normalizing Flows (MNF) [1], but (quoting our response to reviewer3) note several differences as well:
0) Mathematically, MNF treats all the neural net parameters as random variables, and derive a lower-bound on the ELBO to allow for a hierarchical posterior (z->W); whereas we only treat g as random variables, and perform standard variational inference on this model.
1) While the scaling outputs of the normalizing flow in MNF (z) operate on units activations (or equivalently, outgoing weights), in BHNs, the scaling factors (g) operate on the pre-activations (or equivalently, incoming weights).
2) We normalize the direction component of the weights (u := v/||v||), which means that the scale of g (which is analogous to z, in MNF) can control the complexity of the model.  Crucially, this allows us to place a meaningful prior on g, and thus avoid introducing an auxiliary inference model for g.
3) We also perform a more broad experimental evaluation, including experiments on active learning.

It may be worth mentioning that we developed Bayesian Hypernetworks independently, originally for submission to NIPS.
Even in the context of [1], we believe our work provides the following valuable contributions to the community:
1) Further experimental validation that normalizing flows can outperform simpler approaches to variational deep learning.
2) We make connections with hypernetworks and generative models, which might inspire more creative parametrizations and/or applications of generative modelling techniques in this line of work, such as Shi et al. 2017 [2] (although their paper also predates our ICLR submission).
3) While both works demonstrate the benefits of using flexible and powerful approximate posteriors, in contrast to [1], our results suggest that it may not be necessary to construct a full posterior over the weights of a neural network in order to capture these benefits.


We’ve also updated the paper as follows:
1) We’ve corrected the experiments section to explain that our prior is over the scaling factors (g), not the parameters (theta).
2) We’ve added a section to the appendix which gives the derivation of our training objective.
3) We’ve added an additional plot to Figure 1 using a Bayesian Hypernetwork to output a full posterior over all the primary network parameters (theta), as requested by reviewer 1.
4) We’ve fixed the typos pointed out by reviewer 1.

[1] Christos Louizos and Max Welling. Multiplicative normalizing flows for variational bayesian neural
networks. arXiv e-prints, March 2017.
[2] Jiaxin Shi, Shengyang Sun, and Jun Zhu. Implicit variational inference with kernel density ratio
fitting. arXiv preprint arXiv:1705.10119, 2017.

---

### Decision · Program_Chairs · 2018-01-29
**ICLR 2018 Conference Acceptance Decision**

**Decision:**

Reject

**Comment:**

This paper presents a new method for approximate Bayesian inference in neural networks.  The reviewers all found the proposed idea interesting but originally had questions about its novelty (with regard to normalizing flows) and questioned the technical rigor of the approach.  The authors did a good job of addressing the technical concerns, causing two of the reviewers to raise their scores.  However, the paper remains just borderline and none of the reviewers are willing to champion the paper as their questions about novelty and empirical evaluation remain.  The reviewers all questioned fundamental technical aspects of the paper (which were clarified in the discussion), indicating that the paper requires more careful exposition of the technical contributions.  Taking the reviewers feedback and discussion into account, running some more compelling experiments and rewriting the paper to make the technical aspects more clear would make this a much stronger submission.

Pros:
- Provides an interesting idea for approximate Bayesian inference in deep networks
- The paper appears correct
- The approach is scalable and tractable

Cons:
- The technical writing is not rigorous
- The reviewers don't seem convinced by the empirical analysis
- Incremental over existing (but recent) work (Luizos and Welling)